# Deep Unfolding Sparse Bayesian Learning Network for Off-Grid DOA Estimation with Nested Array

Zhenghui Gong, Xiaolong Su *[ID], Panhe Hu, Shuowei Liu [ID] and Zhen Liu [ID]

College of Electronic Science and Technology, National University of Defense Technology,
Changsha 410073, China; gongzhenghui10@nudt.edu.cn (Z.G.); hupanhe13@nudt.edu.cn (P.H.);
liushuowei@nudt.edu.cn (S.L.); zhen_liu@nudt.edu.cn (Z.L.)
* Correspondence: suxiaolong_nudt@163.com

**Abstract:** Recently, deep unfolding networks have been widely used in direction of arrival (DOA) estimation because of their improved estimation accuracy and reduced computational cost. However, few have considered the existence of a nested array (NA) with off-grid DOA estimation. In this study, we present a deep sparse Bayesian learning (DSBL) network to solve this problem. We first establish the signal model for off-grid DOA with NA. Then, we transform the array output into a real domain for neural networks. Finally, we construct and train the DSBL network to determine the on-grid spatial spectrum and off-grid value, where the loss function is calculated using reconstruction error and the sparsity of network output, and the layers correspond to the steps of the sparse Bayesian learning algorithm. We demonstrate that the DSBL network can achieve better generalization ability without training labels and large-scale training data. The simulation results validate the effectiveness of the DSBL network when compared with those of existing methods.

**Keywords:** deep unfolding network; sparse Bayesian learning; off-grid; direction of arrival (DOA) estimation; nested array (NA)

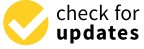



## 1. Introduction

The direction of arrival (DOA) estimation for UAV emitters has been an important application in the field of array signal processing [1–3]. A non-uniform array is an array structure with non-uniform spacing between elements [4,5]. In the case of the same number of elements, the non-uniform array has a larger array aperture than the uniform array, which can improve the resolution of parameter estimation [6–8]. In addition, when the array aperture is the same, a non-uniform array requires fewer elements, which can reduce the hardware cost of the signal processing system and suppress the impact of mutual coupling between elements [9].

The sparse representation method divides the spatial domain into discrete grids, and grid mismatch (GM) occurs when the DOA does not fall on the grid [10], which can reduce the estimation performance of signal parameters. In addition, if the estimation accuracy is improved by reducing the spacing of the grid, the dimensionality of the overcomplete dictionary will lead to an increase in computational complexity in the process of sparse reconstruction. According to sparse reconstruction conditions such as the mutual incoherence property (MIP) and the restricted isometric property (RIP) [11,12], the high correlation between different columns in the overcomplete dictionary with small grid spacing will lead to the failure of sparse reconstruction algorithms. In response to grid mismatch, quantization errors are introduced into the signal model, which does not strictly limit the signal to fall on the grid [13–17]. Yang et al., proposed a mathematical model using basis pursuit denoising (BPDN) to jointly solve the nearest grid and corresponding quantization errors [18]. Compared to the sparse global least squares method [19], the regularization parameters in this method can be set through off-grid mathematical models and noise.

In addition, using the off-grid mathematical model of the first-order Taylor expansion, Yang et al., proposed off-grid sparse Bayesian inference (OGSBI) for off-grid DOA estimation [19], which is suitable for both single and multiple snapshot situations and can reduce computational complexity through singular value decomposition. Moreover, Jagannath et al., analyzed the performance of the off-grid mathematical model and quantization error estimation with the first-order Taylor expansion [20]. Yang et al., developed iterative algorithm with the off-grid model from a Bayesian perspective [21]. Tan proposed a joint sparse recovery method to solve the problem of overcomplete dictionary mismatching, which can improve the accuracy of off-grid parameter estimation [22]. In addition, Wu et al., utilized the perturbation covariance matrix to improve the convergence of sparse Bayesian learning methods for off-grid parameter estimation [23].

The above-mentioned methods can be summarized as model-driven methods [24]. In recent years, deep learning has been gradually applied to DOA estimation [25–27]. However, deep neural networks and convolutional neural networks belong to the black box, and their generalization ability for untrained data is relatively poor. In addition, overfitting may occur during the training process. Noticeably, deep unfolding networks (DUNs) construct the iterative process of the sparse reconstruction method into the hidden layers of networks [28]. Since the hidden layer corresponds to an iterative process of the sparse reconstruction method, a DUN requires fewer layers for convergence than sparse reconstruction methods, which can accelerate DOA estimation. Compared to the traditional deep neural networks and convolutional neural networks, the parameters of the hidden layer in the DUNs have certain mathematical meanings, which correspond to the calculation process of iterative solutions [29,30]. During the training process, the deep unfolding network can learn the regular pattern of data and have generalization ability for untrained samples [31].

Accordingly, the contribution of this work is to construct a deep unfolding network for off-grid DOA estimation with a nested array, which can reduce computational complexity and improve the estimation accuracy. Utilizing the quantization error, we establish a mathematical model for off-grid DOA estimation with the first-order derivative of an overcomplete dictionary. In order to reduce the computational complexity, we transform the complex domain covariance vector of nested arrays into a real domain covariance vector. Considering the dual advantages of model-driven and data-driven methods in deep unfolding networks, we transform the iterative steps of the SBL algorithm into a cascaded form of neural networks and construct a deep SBL network. By alternating the on-grid spatial spectrum and the off-grid quantization error, off-grid angle estimation is achieved. The experimental results show that the computational complexity of the DSBL network is lower than that of the model-driven SBL algorithm. Moreover, the proposed DSBL network can improve the estimation accuracy under a low signal-to-noise ratio.

Notations: Throughout this paper, the italic letters (e.g., *a*), lowercase boldface letters (e.g., **a**), and the capital boldface letters (e.g., **A**) denote variables, vectors, and matrices, respectively. $\| \cdot \|_1$ and $\| \cdot \|_2$ denote $\ell_1$ norm and $\ell_2$ norm, respectively. $\otimes$ and $\odot$ denote Kronecker and Khatri–Rao products, respectively. $E(\cdot)$, $\mathrm{vec}(\cdot)$, and $\mathrm{diag}(\cdot)$ denote mathematical expectation, vectorization operator, and diagonal operator, respectively. $(\cdot)^*$, $(\cdot)^\mathrm{T}$, and $(\cdot)^\mathrm{H}$ denote complex conjugate, transpose, and Hermitian transpose, respectively. $\Re(\cdot)$ and $\Im(\cdot)$ denote the real part and imaginary part of a complex value, respectively.

## 2. Signal Model for Off-Grid DOA with NA

In practice, the geometry of a nested array (NA) contains $M$ elements, where the internal spacing of the first subarray is $d$, which is located at the following position:

$$\{\xi_m | \xi_m = md, m = 1, 2, \ldots, M/2\}, \tag{1}$$

and the internal spacing of the second subarray is $(M/2 + 1)d$, which is located at the following position:

$$\{\xi_m | \xi_m = (m - M/2)(M/2 + 1)d, m = M/2 + 1, M/2 + 2, \ldots, M\}. \tag{2}$$

Considering that the configuration of NAs is impinged by $K$ narrowband signals from different DOAs, the array output at the $n$th snapshot can be expressed as follows:

$$
\begin{aligned}
\boldsymbol{x}(n) &= [x_1(n) x_1(n) \cdots x_M(n)]^{\mathrm{T}} \\
&= \sum_{k=1}^{K} \boldsymbol{a}(\theta_k) s_k(n) + \boldsymbol{w}(n) \\
&= \boldsymbol{A}\boldsymbol{s}(n) + \boldsymbol{w}(n),
\end{aligned}
\tag{3}
$$

where $\mathbf{s}(n) = [s_1(n)\ s_2(n) \cdots s_K(n)]^{\mathrm{T}}$ denotes the vector of $K$ sources at the $n$th snapshot, $\mathbf{A} = [\mathbf{a}(\theta_1)\ \mathbf{a}(\theta_2)\ \cdots \mathbf{a}(\theta_K)]$ stands for the steering matrix, $\mathbf{a}(\theta_k) = [a_1(\theta_k)\ a_2(\theta_k)\cdots a_M(\theta_k)]^{\mathrm{T}}$ with $a_M(\theta_k) = \exp(-\mathrm{j}(2\pi\xi_m \sin\theta_k/\lambda))$ stands for the steering vector of the $k$th signals, $\lambda$ denotes the wavelength of signals, and $\mathbf{w}(n) = [w_1(n)\ w_1(n)\cdots w_M(n)]^{\mathrm{T}}$ denotes the vector of Gaussian white noise at the $n$th snapshot.

We consider the covariance matrix of array output for NAs, which are calculated as follows:

$$
\begin{aligned}
\boldsymbol{R} &= E\left(\mathbf{x}(n)\mathbf{x}^{\mathrm{H}}(n)\right) = \boldsymbol{A}\mathrm{diag}\left(\begin{bmatrix} \sigma_1^2 & \sigma_2^2 & \cdots & \sigma_K^2 \end{bmatrix}^{\mathrm{T}}\right)\boldsymbol{A}^{\mathrm{H}} + \sigma_{\mathrm{w}}^2 \mathbf{I}_M \\
&\approx \frac{1}{N}\sum_{n=1}^{N} \mathbf{x}(n)\mathbf{x}^{\mathrm{H}}(n),
\end{aligned}
\tag{4}
$$

where $\sigma_k^2$ denotes the power of the $k$th signal and $\mathbf{I}_M$ denotes the $M \times M$ dimensional identity matrix.

Therefore, the vector form of the covariance matrix can be expressed as follows:

$$
\begin{aligned}
\boldsymbol{y} &= \mathrm{vec}(\boldsymbol{R}) \\
&= (\boldsymbol{A}^* \odot \boldsymbol{A})[\sigma_1^2\ \sigma_2^2 \cdots \sigma_K^2]^{\mathrm{T}} + \sigma_{\mathrm{W}}^2[\boldsymbol{\eta}_1^{\mathrm{T}}\ \boldsymbol{\eta}_2^{\mathrm{T}} \cdots \boldsymbol{\eta}_M^{\mathrm{T}}]^{\mathrm{T}},
\end{aligned}
\tag{5}
$$

where $\boldsymbol{\eta}_m^{\mathrm{T}}$ denotes the $M$ dimensional vector, the $m$th element of $\boldsymbol{\eta}_m^{\mathrm{T}}$ is 1, and the remaining elements are 0; the equivalent steering matrix $\boldsymbol{A}^* \odot \boldsymbol{A}$ can be calculated as follows:

$$\boldsymbol{A}^* \odot \boldsymbol{A} = [\boldsymbol{a}^*(\theta_1) \otimes \boldsymbol{a}(\theta_1)\boldsymbol{a}^*(\theta_2) \otimes \boldsymbol{a}(\theta_2) \cdots \boldsymbol{a}^*(\theta_K) \otimes \boldsymbol{a}(\theta_K)], \tag{6}$$

The scenario of off-grid DOA estimation is shown in Figure 1. In the off-grid case, using the first-order derivative of the overcomplete dictionary on the grid, the sparse representation of the covariance vector in (5) can be constructed as follows:

$$\boldsymbol{y} = (\boldsymbol{\Phi} + \boldsymbol{FB})\boldsymbol{z} + \sigma_{\mathrm{w}}^2[\boldsymbol{\eta}_1^{\mathrm{T}}\ \boldsymbol{\eta}_2^{\mathrm{T}} \cdots \boldsymbol{\eta}_M^{\mathrm{T}}]^{\mathrm{T}}, \tag{7}$$

where $\sigma_{\mathrm{w}}^2$ denotes the noise power, $\boldsymbol{z} = [z_1\ z_2\ \cdots\ z_Q]^{\mathrm{T}}$ denotes the spatial spectrum on the grid, and the values of $\{\theta_{q_1}, \theta_{q_2}, \ldots, \theta_{q_K}\}$ in the spatial spectrum are $\{\sigma_1^2, \sigma_2^2, \cdots, \sigma_K^2\}$. In addition, $\boldsymbol{\Phi} + \boldsymbol{FB}$ denotes the off-grid overcomplete dictionary, and $\boldsymbol{\Phi}$ denotes the on-grid overcomplete dictionary, which can be expressed as follows:

$$
\begin{aligned}
\boldsymbol{\Phi} &= [\boldsymbol{\varphi}(\theta_1)\ \boldsymbol{\varphi}(\theta_2) \cdots \boldsymbol{\varphi}(\theta_Q)] \\
&= [\boldsymbol{a}^*(\theta_1) \otimes \boldsymbol{a}(\theta_1)\quad \boldsymbol{a}^*(\theta_2) \otimes \boldsymbol{a}(\theta_2) \cdots \boldsymbol{a}^*(\theta_Q) \otimes \boldsymbol{a}(\theta_Q)]
\end{aligned}
\tag{8}
$$

where $\boldsymbol{a}(\theta_q) = [a_1(\theta_q)\ a_2(\theta_q)\cdots a_M(\theta_q)]^{\mathrm{T}}$ denotes the steering vector corresponding to the $q$th angle in the angle set $\{\theta_1, \theta_2, \ldots, \theta_Q\}$ and $\boldsymbol{F}$ denotes the first derivative of the overcomplete dictionary on the grid, which can be expressed as follows:

$$F = \begin{bmatrix} f(\theta_1) \, f(\theta_2) \cdots f(\theta_Q) \end{bmatrix}$$
$$= \begin{bmatrix} \boldsymbol{\varphi}'(\theta_1) \; \boldsymbol{\varphi}'(\theta_2) \cdots \boldsymbol{\varphi}'(\theta_Q) \end{bmatrix}$$
$$= \begin{bmatrix} (a^*(\theta_1) \otimes a(\theta_1))' \; (a^*(\theta_2) \otimes a(\theta_2))' \cdots (a^*(\theta_Q) \otimes a(\theta_Q))' \end{bmatrix} \tag{9}$$

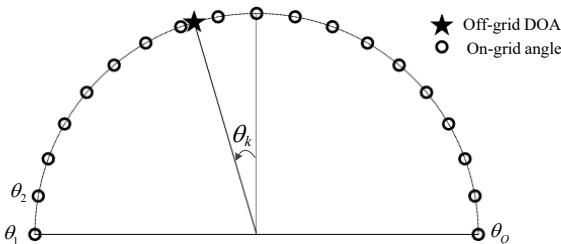

**Figure 1.** The scenario of off-grid DOA estimation.

In addition, $B$ denotes the quantization error matrix, which can be expressed as the following:

$$B = \mathrm{diag}(b) \tag{10}$$

where $b = [b_1 \, b_2 \cdots b_Q]^{\mathrm{T}}$ represents the quantization error vector, the values of $\{\theta_{q_1}, \theta_{q_2}, \ldots, \theta_{q_K}\}$ in the quantization error vector are $\{\Delta_1, \Delta_2, \cdots, \Delta_K\}$, and the values of the remaining elements in the quantization error vector are 0; $\Delta_{q_k} = \theta_k - \theta_{q_k}$ denotes the quantization error between the DOA of the $k$th signal and the closest angle on the grid.

## 3. Proposed Algorithm

In this section, the DSBL network is used to determine the on-grid spatial spectrum and off-grid quantization error via NA, where the layers of the DSBL network correspond to the steps of the model-driven SBL method. Since the neural networks are suitable for dealing with the real-valued data, we transformed the array output into the real domain in advance.

### 3.1. Transformation of the Array Output to the Real Domain

Considering that the neural networks are suitable for dealing with real-valued data, we rewrote Equation (7) to the following form:

$$\begin{bmatrix} \Re(\mathbf{y}) \\ \Im(\mathbf{y}) \end{bmatrix} = \begin{bmatrix} \Re(\boldsymbol{\Phi} + FB) & -\Im(\boldsymbol{\Phi} + FB) \\ \Im(\boldsymbol{\Phi} + FB) & \Re(\boldsymbol{\Phi} + FB) \end{bmatrix} \begin{bmatrix} \Re(\mathbf{z}) \\ \Im(\mathbf{z}) \end{bmatrix} + \begin{bmatrix} \Re\left(\sigma_{\mathrm{w}}^2 [\boldsymbol{\eta}_1^{\mathrm{T}} \;\; \boldsymbol{\eta}_2^{\mathrm{T}} \;\; \cdots \;\; \boldsymbol{\eta}_M^{\mathrm{T}}]^{\mathrm{T}}\right) \\ \Im\left(\sigma_{\mathrm{w}}^2 [\boldsymbol{\eta}_1^{\mathrm{T}} \;\; \boldsymbol{\eta}_2^{\mathrm{T}} \;\; \cdots \;\; \boldsymbol{\eta}_M^{\mathrm{T}}]^{\mathrm{T}}\right) \end{bmatrix}. \tag{11}$$

Since $z$ is considered as the DOA spatial spectrum with source power, and its imaginary part is zero, Equation (11) can be equivalently rewritten as follows:

$$\begin{bmatrix} \Re(\mathbf{y}) \\ \Im(\mathbf{y}) \end{bmatrix} = \begin{bmatrix} \Re(\boldsymbol{\Phi} + FB) \\ \Im(\boldsymbol{\Phi} + FB) \end{bmatrix} z + \begin{bmatrix} \sigma_{\mathrm{w}}^2 [\boldsymbol{\eta}_1^{\mathrm{T}} \, \boldsymbol{\eta}_2^{\mathrm{T}} \cdots \boldsymbol{\eta}_M^{\mathrm{T}}]^{\mathrm{T}} \\ \mathbf{0}_{M^2 \times 1} \end{bmatrix}$$
$$= \left( \begin{bmatrix} \Re(\boldsymbol{\Phi}) \\ \Im(\boldsymbol{\Phi}) \end{bmatrix} + \begin{bmatrix} \Re(F) \\ \Im(F) \end{bmatrix} B \right) z + \begin{bmatrix} \sigma_{\mathrm{w}}^2 [\boldsymbol{\eta}_1^{\mathrm{T}} \, \boldsymbol{\eta}_2^{\mathrm{T}} \cdots \boldsymbol{\eta}_M^{\mathrm{T}}]^{\mathrm{T}} \\ \mathbf{0}_{M^2 \times 1} \end{bmatrix} \tag{12}$$

where

$$\Re(\boldsymbol{\Phi}) = \begin{bmatrix} \Re(\boldsymbol{\varphi}(\theta_1)) \; \Re(\boldsymbol{\varphi}(\theta_2)) \cdots \Re(\boldsymbol{\varphi}(\theta_q)) \cdots \Re(\boldsymbol{\varphi}(\theta_Q)) \end{bmatrix} \tag{13}$$

$$\Im(\boldsymbol{\Phi}) = \begin{bmatrix} \Im(\boldsymbol{\varphi}(\theta_1)) \; \Im(\boldsymbol{\varphi}(\theta_2)) \cdots \Im(\boldsymbol{\varphi}(\theta_q)) \cdots \Im(\boldsymbol{\varphi}(\theta_Q)) \end{bmatrix} \tag{14}$$

and $\boldsymbol{\varphi}(\theta_q)$ denotes the $q$th column of $\boldsymbol{\Phi}$. Furthermore, $\Re(F)$ and $\Im(F)$ can be expressed as follows:

$$\Re(F) = \begin{bmatrix} \Re(f(\theta_1)) \; \Re(f(\theta_2)) \cdots \Re(f(\theta_q)) \cdots \Re(f(\theta_Q)) \end{bmatrix} \tag{15}$$

$$\Im(\boldsymbol{F}) = \left[\Im(\boldsymbol{f}(\theta_1))\ \Im(\boldsymbol{f}(\theta_2))\cdots \Im(\boldsymbol{f}(\theta_q))\cdots \Im(\boldsymbol{f}(\theta_Q))\right] \tag{16}$$

where $\boldsymbol{f}(\theta_q)$ denotes the $q$th column of $\boldsymbol{F}$.

### 3.2. Deep Unfolding Sparse Bayesian Learning Network

In order to accelerate the convergence speed of the SBL algorithm, we expanded the iterative steps of the SBL method into the network of cascade form, where the estimation of off-grid DOA is calculated by the peaks of the on-grid spatial spectrum and the corresponding off-grid quantization errors. As shown in Figure 2, the DSBL network contains $L$ layers for on-grid spatial spectrum estimation and off-grid quantization error estimation, where the previous layer of off-grid quantization error matrix $\boldsymbol{B}$ is used to estimate the current layer of on-grid spatial spectrum $\boldsymbol{Z}$, and the current layer of on-grid spatial spectrum $\boldsymbol{Z}$ is used to estimate the previous layer of off-grid quantization error matrix $\boldsymbol{B}$.

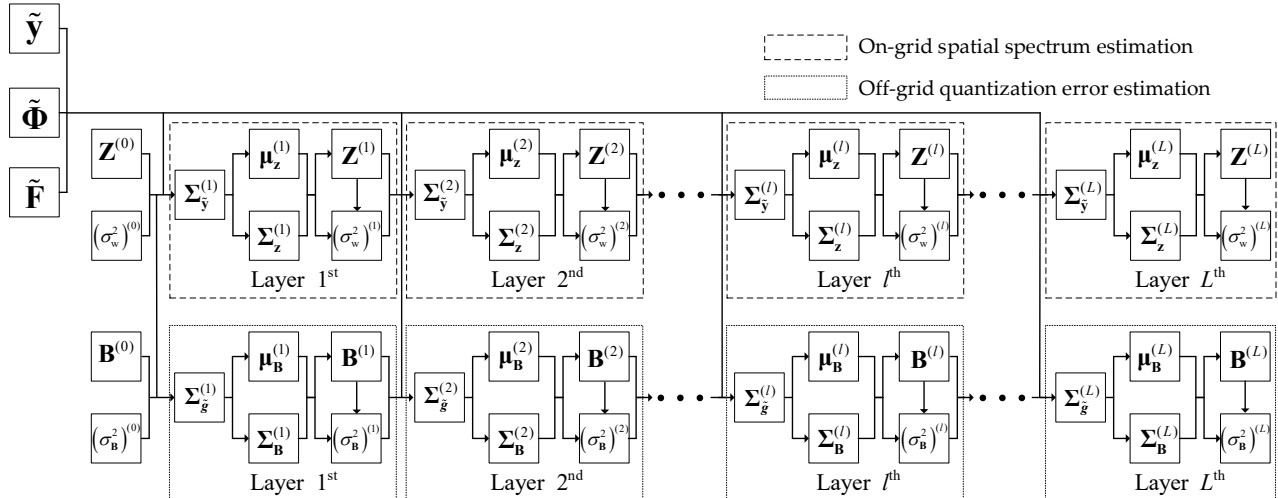

**Figure 2.** Scheme of the proposed DSBL network for off-grid DOA estimation.

In practice, the covariance vector in the real domain can be expressed as follows:

$$\widetilde{\boldsymbol{y}} = \left(\widetilde{\boldsymbol{\Phi}} + \widetilde{\boldsymbol{F}}\boldsymbol{B}\right)\boldsymbol{z} + \begin{bmatrix} \sigma_{\mathrm{w}}^2 [\boldsymbol{\eta}_1^{\mathrm{T}}\ \boldsymbol{\eta}_2^{\mathrm{T}}\ \cdots\ \boldsymbol{\eta}_M^{\mathrm{T}}]^{\mathrm{T}} \\ \boldsymbol{0}_{M^2 \times 1} \end{bmatrix} \tag{17}$$

where $\widetilde{\boldsymbol{y}} = [\Re(\boldsymbol{y}^{\mathrm{T}})\ \Im(\boldsymbol{y}^{\mathrm{T}})]^{\mathrm{T}}$, $\widetilde{\boldsymbol{\Phi}} = [\Re(\boldsymbol{\Phi}^{\mathrm{T}})\ \Im(\boldsymbol{\Phi}^{\mathrm{T}})]^{\mathrm{T}}$, and $\widetilde{\boldsymbol{F}} = [\Re(\boldsymbol{F})^{\mathrm{T}}\ \Im(\boldsymbol{F})^{\mathrm{T}}]^{\mathrm{T}}$.

Considering that the noise part in Equation (17) makes the optimization problem more nebulous [32], we constructed the following convex relaxation:

$$\min\left( \left\| \widetilde{\boldsymbol{y}} - \left(\widetilde{\boldsymbol{\Phi}} + \widetilde{\boldsymbol{F}}\boldsymbol{B}\right)\boldsymbol{z} \right\|_2^2 + \zeta\|\boldsymbol{z}\|_1 \right). \tag{18}$$

By integrating the amplitudes of spatial power in the SBL framework [33], the probability of $\widetilde{\boldsymbol{y}}$ with respect to the hyperparameters $\boldsymbol{z}$ and $\sigma_{\mathrm{w}}^2$ can be expressed as follows:

$$\begin{aligned} p\left(\widetilde{\boldsymbol{y}}|\boldsymbol{z},\sigma_{\mathrm{w}}^2\right) &= \int p\left(\widetilde{\boldsymbol{y}}|\boldsymbol{z},\sigma_{\mathrm{w}}^2\right)p(\boldsymbol{z}|\boldsymbol{\gamma})d\boldsymbol{z} \\ &= \frac{\exp\left(-\mathrm{tr}\left(\widetilde{\boldsymbol{y}}^{\mathrm{H}}\boldsymbol{\Sigma}_{\widetilde{\boldsymbol{y}}}^{-1}\widetilde{\boldsymbol{y}}\right)\right)}{\det(\pi\boldsymbol{\Sigma}_{\widetilde{\boldsymbol{y}}})}, \end{aligned} \tag{19}$$

where $\boldsymbol{z} = [z_1\ z_2\cdots z_Q]^{\mathrm{T}}$ denotes the on-grid spatial power spectrum and $\boldsymbol{\Sigma}_{\widetilde{\boldsymbol{y}}} = \left(\widetilde{\boldsymbol{\Phi}} + \widetilde{\boldsymbol{F}}\boldsymbol{B}\right)\boldsymbol{Z}\left(\widetilde{\boldsymbol{\Phi}} + \widetilde{\boldsymbol{F}}\boldsymbol{B}\right)^{\mathrm{H}} + \sigma_{\mathrm{w}}^2\mathbf{I}_{2M^2}$ with $\boldsymbol{Z} = \mathrm{diag}(\boldsymbol{z})$ denotes the covariance matrix of the array output.

Therefore, the hyperparameters $z$ and $\sigma_w^2$ can be estimated by maximizing $p(\widetilde{y}|z, \sigma_w^2)$, which can be considered as the type-II maximum likelihood (ML) problem and is derived from the EM method in [34]. In this study, $z$ and $\sigma_w^2$ are updated by exploiting the iterative E-steps and M-steps of the EM method.

The estimation of the on-grid spatial spectrum in the DSBL network consists of $L$ layers, where $\widetilde{y}$, $\widetilde{\Phi}$, and $\widetilde{F}$ are considered as the input of each layer, and the initial hyperparameters are set as $Z^{(0)} = \mathrm{eye}(Q)$ and $(\sigma_w^2)^{(0)} = 1$. Based on the EM method, the E-step in the $l$th layer for on-grid spatial spectrum estimation is performed to calculate the posteriori mean and posteriori covariance:

$$\boldsymbol{\mu}_z^{(l)} = Z^{(l-1)}\left(\widetilde{\Phi} + \widetilde{F}B\right)^{\mathrm{H}}(\boldsymbol{\Sigma}_{\widetilde{y}}^{(l-1)})^{-1}\widetilde{y}, \tag{20}$$

$$\boldsymbol{\Sigma}_z^{(l)} = Z^{(l-1)} - Z^{(l-1)}\left(\widetilde{\Phi} + \widetilde{F}B\right)^{\mathrm{H}}(\boldsymbol{\Sigma}_{\widetilde{y}}^{(l-1)})^{-1}\left(\widetilde{\Phi} + \widetilde{F}B\right)Z^{(l-1)}, \tag{21}$$

for $l = 1, 2, \ldots, L$, where $\boldsymbol{\Sigma}_{\widetilde{y}}^{(l-1)} = \left(\widetilde{\Phi} + \widetilde{F}B\right)Z^{(l-1)}\left(\widetilde{\Phi} + \widetilde{F}B\right)^{\mathrm{H}} + (\sigma_w^2)^{(l-1)}I_{2M^2}$ denotes the array covariance.

Moreover, the M-step in the $l$th layer for on-grid spatial spectrum estimation is performed to calculate the following:

$$Z^{(l)} = \boldsymbol{\mu}_z^{(l)}(\boldsymbol{\mu}_z^{(l)})^{\mathrm{T}} + \boldsymbol{\Sigma}_z^{(l)}, \tag{22}$$

and the corresponding noise variance for on-grid spatial spectrum estimation is derived from the following:

$$(\sigma_w^2)^{(l)} = \frac{1}{2M^2}\left\|\widetilde{y} - \left(\widetilde{\Phi} + \widetilde{F}B\right)\boldsymbol{\mu}_z^{(l)}\right\|_2^2 + \frac{(\sigma_w^2)^{(l-1)}}{2M^2}\left(Q - \sum_{q=1}^{Q}\frac{(\boldsymbol{\Sigma}_z^{(l)})_{q,q}}{(Z^{(l)})_{q,q}}\right), \tag{23}$$

where $(\boldsymbol{\Sigma}_z^{(l)})_{q,q}$ and $(Z^{(l)})_{q,q}$ denote the $(q,q)$th element of $\boldsymbol{\Sigma}_z^{(l)}$ and $Z^{(l)}$, respectively.

As for the estimation of the off-grid quantization error matrix $B$, the sparse representation of the covariance vector in the real domain can be constructed as follows:

$$\widetilde{y} - \widetilde{\Phi}z = \widetilde{F}(Bz) + \begin{bmatrix} \sigma_w^2[\boldsymbol{\eta}_1^{\mathrm{T}} \, \boldsymbol{\eta}_2^{\mathrm{T}} \cdots \boldsymbol{\eta}_M^{\mathrm{T}}]^{\mathrm{T}} \\ \mathbf{0}_{M^2 \times 1} \end{bmatrix}, \tag{24}$$

Similarly, the optimization problem of off-grid quantization error estimation can be expressed as follows:

$$\min\left(\left\|\widetilde{g} - \widetilde{F}(Bz)\right\|_2^2 + \zeta_2\|Bz\|_1\right), \tag{25}$$

where $\widetilde{g} = \widetilde{y} - \widetilde{\Phi}z$. In this study, the estimation of the off-grid quantization error in the DSBL network consists of $L$ layers; the initial hyperparameters are set as $B^{(0)} = \mathrm{eye}(Q)$ and $(\sigma_B^2)^{(0)} = 1$.

Based on the EM method, the E-step in the $l$th layer for off-grid quantization error estimation is performed to calculate the posteriori mean and posteriori covariance:

$$\boldsymbol{\mu}_B^{(l)} = \boldsymbol{\Gamma}^{(l-1)}\widetilde{F}^{\mathrm{H}}(\boldsymbol{\Sigma}_{\widetilde{g}}^{(l-1)})^{-1}\widetilde{g}^{(l-1)}, \tag{26}$$

$$\boldsymbol{\Sigma}_B^{(l)} = \boldsymbol{\Gamma}^{(l-1)} - \boldsymbol{\Gamma}^{(l-1)}\widetilde{F}^{\mathrm{H}}(\boldsymbol{\Sigma}_{\widetilde{g}}^{(l-1)})^{-1}\widetilde{F}\boldsymbol{\Gamma}^{(l-1)}, \tag{27}$$

for $l = 1, 2, \ldots, L$, where $\widetilde{g}^{(l-1)} = \widetilde{y} - \widetilde{\Phi}\mathrm{diag}(Z^{(l-1)})$, and $\boldsymbol{\Sigma}_{\widetilde{g}}^{(l-1)} = \widetilde{F}\boldsymbol{\Gamma}^{(l-1)}\widetilde{F}^{\mathrm{H}} + (\sigma_B^2)^{(l-1)}I_{2M^2}$ denotes the array covariance.

Moreover, the *M*-step in the *l*th layer of the off-grid quantization error estimation is performed to calculate the following:

$$\boldsymbol{\Gamma}^{(l)} = \boldsymbol{\mu}_{\boldsymbol{B}}^{(l)} (\boldsymbol{\mu}_{\boldsymbol{B}}^{(l)})^{\mathrm{T}} + \boldsymbol{\Sigma}_{\boldsymbol{B}}^{(l)}, \tag{28}$$

and the corresponding noise variance of the off-grid quantization error estimation is derived from the following:

$$(\sigma_{\boldsymbol{B}}^2)^{(l)} = \frac{1}{2M^2} \left\| \widetilde{\boldsymbol{g}} - \widetilde{\boldsymbol{F}} \boldsymbol{\mu}_{\boldsymbol{B}}^{(l)} \right\|_2^2 + \frac{(\sigma_{\boldsymbol{B}}^2)^{(l-1)}}{2M^2} \left( Q - \sum_{q=1}^{Q} \frac{(\boldsymbol{\Sigma}_{\boldsymbol{B}}^{(l)})_{q,q}}{(\boldsymbol{\Gamma}^{(l)})_{q,q}} \right), \tag{29}$$

where $(\boldsymbol{\Sigma}_{\boldsymbol{B}}^{(l)})_{q,q}$ and $(\boldsymbol{\Gamma}^{(l)})_{q,q}$ denote the (*q*,*q*)th element of $\boldsymbol{\Sigma}_{\boldsymbol{B}}^{(l)}$ and $\boldsymbol{\Gamma}^{(l)}$, respectively.

Therefore, by employing the output of the *l*th layer for the on-grid spatial spectrum $\boldsymbol{Z}^{(l)}$ and off-grid quantization error $\boldsymbol{\Gamma}^{(l)}$, the *q*th element on the diagonal of the off-grid quantization error matrix $\boldsymbol{B}^{(l)}$ can be calculated as follows:

$$b_q^{(l)} = \Gamma_q^{(l)} / z_q^{(l)}, \tag{30}$$

Generally, in the training progress of the proposed DSBL network, a stochastic gradient descent (SGD) is exploited to renew the trainable parameters. Referring to the convex relaxation in Equations (18) and (25), the loss function is defined as follows:

$$\min \left( \sum_{t=1}^{T} \left( \left\| \widetilde{\mathbf{y}}_t - \left( \widetilde{\boldsymbol{\Phi}} + \widetilde{\boldsymbol{F}} \boldsymbol{B} \right) \mathrm{diag}(\boldsymbol{Z}_t^{(L)}) \right\|_2^2 + \zeta_1 \left\| \mathrm{diag}(\boldsymbol{Z}_t^{(L)}) \right\|_1 + \zeta_2 \left\| \boldsymbol{B}_t^{(L)} \mathrm{diag}(\boldsymbol{Z}_t^{(L)}) \right\|_1 \right) \right). \tag{31}$$

for *t* = 1, 2, . . ., *T*, where *T* stands for the total number of samples in the dataset, $\widetilde{\mathbf{y}}_t$ stands for the network input, and $\boldsymbol{Z}_t^{(L)}$ and $\boldsymbol{B}_t^{(L)}$ stand for the estimated DOA spectrum from the network output. Moreover, the proposed DSBL network can determine the off-grid DOA without training labels and large-scale training data, which have generalization abilities with interpretable parameters and layers for off-grid DOA estimation.

Based on the output of the *l*th layer of the on-grid spatial spectrum estimation and the output of the *l*th layer of the off-grid quantization error estimation, the off-grid DOA of the *k*th signal can be calculated as follows:

$$\hat{\theta}_k = \theta_k^{(L)} + b_k^{(L)} \tag{32}$$

where $\theta_k^{(L)}$ stands for the angle corresponding to the *k*th spectral peak in $z^{(L)}$ and $b_k^{(L)}$ stands for the value of the *k*th spectral peak in $\boldsymbol{b}^{(L)}$, where $\boldsymbol{b}^{(L)} = [b_1^{(L)} \ b_2^{(L)} \cdots b_Q^{(L)}]^{\mathrm{T}}$.

### 3.3. Network Implementation of Proposed Method

Overall, the main steps of the trained DSBL network for DOA estimation are summarized in Algorithm 1 (The process of implementing a proposed DSBL algorithm).

---

**Algorithm 1** DSBL Network for Off-Grid DOA Estimation with NA

---

1:    Calculate the covariance matrix using Equation (4).
2:    Apply the vector form of covariance matrix in Equation (5).
3:    Combine real and imaginary parts in Equation (12) as the input of the DSBL network.
4:    Perform the trained DSBL network to acquire spatial spectrum and off-grid quantization error.
5:    Obtain off-grid DOA from the peaks of the spatial spectrum and the corresponding off-grid quantization error in Equation (3).

---

As for the computational complexity of the proposed DSBL network, the covariance matrix in Equation (4) requires $M^2N$ multiplications and $M^2(N-1)$ additions; the vectorization in Equation (5) and combination in Equation (12) do not require additional calculation. When employing the trained deep unfolded SBL network to obtain the on-grid spatial spectrum and off-grid quantization error, the calculation of $\Sigma_{\underline{\textbf{y}}}^{(l)}$ and $\Sigma_{\underline{\textbf{g}}}^{(l)}$ requires $4M^4Q + 2M^2Q$ multiplications and $4M^4(Q-1)$ additions; the calculation of $\mu_{\textbf{z}}^{(l)}$ and $\mu_{B}^{(l)}$ requires $4QM^2(M^2+1)$ multiplications, $Q(4M^4-1)$ additions, and $O(2M^2)$ for the inverse operator; the calculation of $\Sigma_{\textbf{z}}^{(l)}$ and $\Sigma_{B}^{(l)}$ requires $Q(4M^4+2M^2+2M^2Q+Q^2)$ multiplications, $Q(4M^4-2M^2+2M^2Q+Q^2-2Q)$ additions, $2M^2$ subtractions, and $O(2M^2)$ for the inverse operator; the calculation of $\textbf{Z}^{(l)}$ and $B^{(l)}$ requires $Q^2$ multiplications and $2Q^2$ additions; and the calculation of $(\sigma_{\text{w}}^2)^{(l)}$ in Equation (23) requires $2M^2(Q+1)$ multiplications, $2QM^2+Q-2$ additions, $2M^2$ subtractions, and $Q+2$ divisions.

## 4. Computer Simulation Experiments

In this section, a two-level NA with six sensors was exploited to investigate the performance of the DSBL network for off-grid DOA estimation. Specifically, the locations were set as [1,2,3,4,8,12]$d$, and the angle interval of the overcomplete dictionary was set as 1°. Of the samples, 80% of those in the dataset were used for network training, and 20% were used for network validation. Each sample was generated by two signals, with an off-grid angle between −60° and 60°. The signal-to-noise ratio (SNR) was selected from 0 dB to 20 dB, and the number of snapshots was selected from 100 to 500. During the training process of the network, the batch size, epoch, and learning rate were set to 16, 16, and 0.01, respectively.

### 4.1. Layer Number Determination

In this subsection, we employed simulation experiments to determine the layer number. During the training process of the DSBL network, the appropriate layer number was determined by the mean square error (MSE). In this research, the MSE is defined as follows:

$$\text{MSE} = \frac{1}{T}\sum_{t=1}^{T}\left((\textbf{Z}_t^{(L)} - \textbf{Z}_t^{\text{label}}) + (\textbf{B}_t^{(L)} - \textbf{B}_t^{\text{label}})\right)^2. \tag{33}$$

where $\textbf{Z}_t^{(L)}$ and $\textbf{Z}_t^{\text{label}}$ denote the output and label of the on-grid spatial spectrum for the $t$th data, respectively; $\textbf{B}_t^{(L)}$ and $\textbf{B}_t^{\text{label}}$ denote the output and label of the off-grid quantization error matrix for the $t$th data, respectively.

During the training and validation process of the DSBL network, the variation in the RMSE with the epoch is shown in Figure 3. From the figure, it can be seen that the RMSEs of layers 10, 20, 30, and 40 gradually decrease with the increase in epoch during the training process, and the RMSE of the 40-layer network is smaller than that of the other layer networks. This indicates that the estimation accuracy of the 40-layer network is better than that of the other layer networks. Due to insufficient training in the initial stage, when the epoch is less than five, the RMSEs of the networks with more layers are greater than those of networks with fewer layers. As the epoch increases, the RMSEs of networks with more layers gradually decrease compared to networks with fewer layers. After the training process of the DSBL network is completed, the RMSE of the 40-layer network is slightly smaller than that of the 30-layer network. In order to balance the accuracy of the off-grid angle estimation and computational complexity, the DSBL network is set to 30 layers. In addition, as shown in Figure 3b, during the validation process, the RMSEs of the 10-, 20-, 30-, and 40-layer networks gradually decreased with the increase in epoch, indicating that there was no overfitting during the training process of the DSBL network.

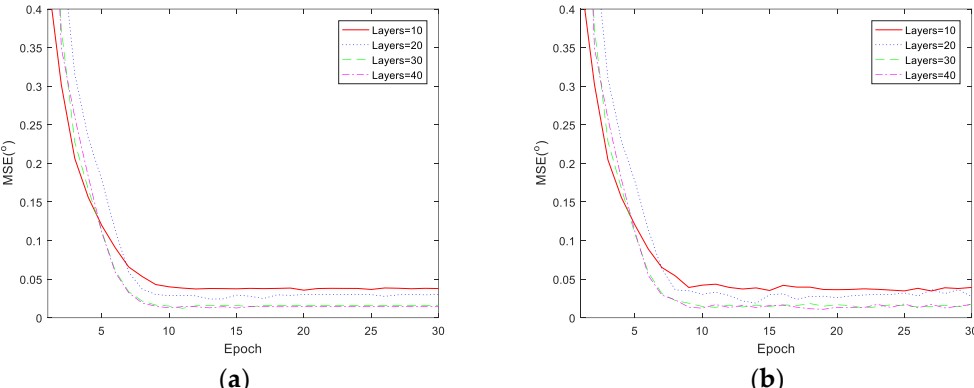

**Figure 3.** MSEs of different layer numbers. (**a**) Training; (**b**) validation.

### 4.2. Comparison of Convergence Performance

When the off-grid DOA of the test samples are set to $-10.95°$ and $2.98°$, Figure 4 shows the relative error of the off-grid DOA estimation. The red solid line represents the relative error of the DSBL network, and the blue dashed line represents the relative error of the model-driven algorithm. As shown in Figure 4, compared to the 8-layer DSBL network, the model-driven algorithm converges after 16 iterations. Due to the computational complexity of each layer in the DSBL network being the same as in the model-driven algorithm, the DSBL network can converge in a shorter time.

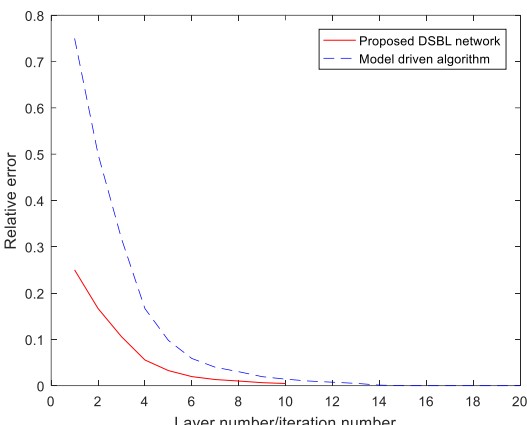

**Figure 4.** Comparison of relative errors.

### 4.3. Generalization Ability for Off-Grid DOA Estimation

In order to verify the generalization ability of the deep unfolding network for off-grid DOAs under different numerical conditions, a total of 120 test samples were generated, with each containing a signal. The angle of the spatial spectrum on the grid was set to $-60°$ to $59°$ with an interval of $1°$; the off-grid quantization error was set to a random value of $0°$ to $1°$; and the signal-to-noise ratio was set to 5 dB. The off-grid DOA estimates obtained through the DSBL network are shown in Figure 5a, and the off-grid estimation error is shown in Figure 5b. It can be seen that the DSBL network has the ability to generalize the off-grid angle under different numerical conditions. Due to the fact that the DSBL network models the iterative steps of the corresponding sparse reconstruction algorithm as hidden layers of the neural network, the network parameters have certain mathematical meanings. During the training process, the deep unfolding network can learn the rules hidden behind the data. Therefore, for untrained data, the DSBL network can also estimate the off-grid angle.

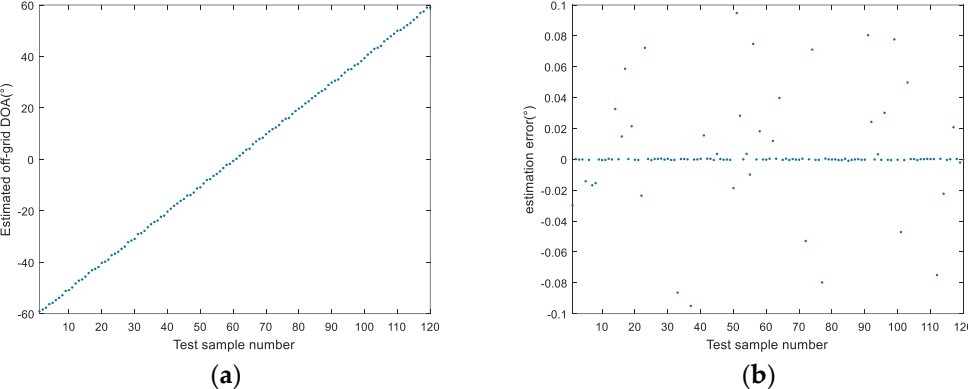

**Figure 5.** DOA estimation of two off-grid sources. (**a**) DOA estimates; (**b**) estimation error.

In this study, the DOA estimates and errors are shown in Figure 5a,b, respectively, where the abscissa denotes the testing index and the red dots and blue dots denote the first sources and the second sources, respectively. Therefore, we can conclude that the trained DSBL network has a generalization ability for the DOA estimation of off-grid sources.

### 4.4. RMSE Comparison of DOA Estimation

In this subsection, root mean square error (RMSE) analysis is performed to investigate the performance of the proposed DSBL network. In this study, the RMSE of DOA estimation is defined as follows:

$$\text{RMSE}(\theta) = \sqrt{\frac{1}{VK}\sum_{v=1}^{V}\sum_{k=1}^{K}\left(\hat{\theta}_k^{(v)} - \theta_k\right)^2}, \tag{34}$$

where $\theta_k$ denotes the real DOA of the $k$th source and $\hat{\theta}_k^{(v)}$ denotes the estimated DOA of the $k$th source in the $v$th Monte Carlo simulation experiment.

The RMSE of the proposed DSBL network was compared with the FOCUSS network in [35], the RVSBL algorithm in [36], the JSR algorithm in [13], and the Cramér–Rao lower bound (CRLB) in [37]. In total, 500 simulation experiments were performed to calculate the RMSEs of two sources, where the DOAs were set as $-10.1°$ and $20.8°$, respectively. The RMSE versus the SNR and snapshot number are shown in Figure 6a,b, respectively, where the RMSE gradually decreases with the increase in the SNR and snapshot number. Since the noise power is not updated in the iterative process of the JSR algorithm and RVSBL algorithm, the accuracy of the DSBL network outperforms the existing methods.

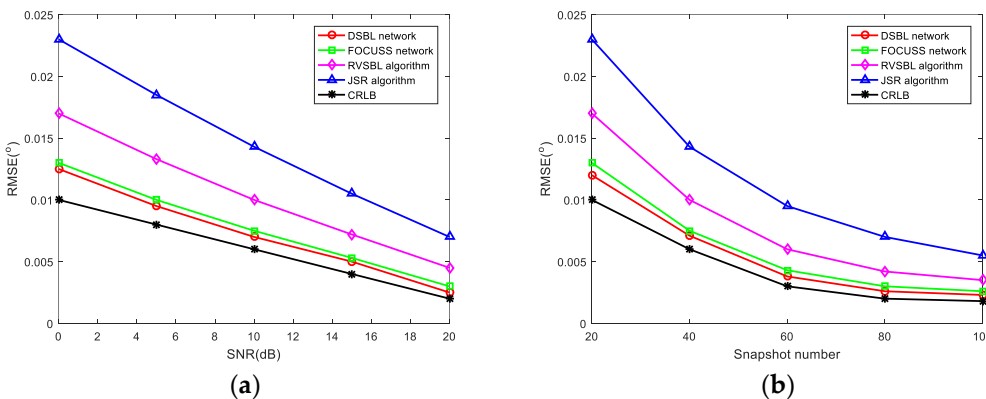

**Figure 6.** RMSEs of different methods. (**a**) SNR; (**b**) snapshot.

### 5. Conclusions

In this research, the DSBL network was constructed for off-grid DOA estimation using the geometry of NAs with mutual coupling. Firstly, the array covariance of the NA was

transformed into an equivalent single snapshot, which can form a continuous array with virtual sensors and increase the degrees of freedom. Then, the vectorization of the array covariance was transformed into the real domain and considered as the input to the DSBL network. Next, the DSBL network was constructed and trained to determine the MCC, where the iterative steps of the EM algorithm were transformed into the layers of the DSBL network, and the loss function was only related to the reconstruction error and the sparsity of the network output. Therefore, the training labels and large-scale training data were not required during the training process of the DSBL network. Finally, the off-grid DOA can be obtained from the peaks of the spatial spectrum and the corresponding off-grid quantization error. The simulation results demonstrate that the proposed DSBL network has better generalization ability with interpretable parameters and layers for off-grid DOA estimation with different source numbers. Compared with the joint sparse recovery method, the SBL method, and the RARE method, the proposed DSBL network achieved a more accurate DOA estimation in the cases of limited snapshot numbers and low SNRs.

**Author Contributions:** Conceptualization, Z.G. and X.S.; methodology, Z.G. and P.H.; software, X.S. and S.L.; writing—review and editing, Z.G. and X.S.; supervision, Z.L. All authors have read and agreed to the published version of the manuscript.

**Funding:** This research was supported in part by the National Natural Science Foundation of China (62201588, 62022091, 61921001) and, in part, by the research program of the National University of Defense Technology (ZK21-14).

**Data Availability Statement:** Data sharing not applicable. No new data were created or analyzed in this study. Data sharing is not applicable to this article.

**Conflicts of Interest:** The authors declare there are no conflict of interest.

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
