# Peer review of "Deep Unfolding Sparse Bayesian Learning Network for Off-Grid DOA Estimation with Nested Array"

_remotesensing, doi:10.3390/rs15225320_

Round 1

Reviewer 1 Report

Comments and Suggestions for Authors

I have the following comments:

In the introduction section, the authors should explicitly incorporate their work's primary contribution.

The introduction needs improvements in clarity and comprehensibility, especially for readers who may need a background in DOA estimation, non-uniform arrays, and related signal processing concepts. Additional explanations, definitions, and context would make it more accessible to a broader audience.

The research design could include:

  1. It is essential to ensure that experiments are well-controlled. To enhance the reliability of the results, the authors can consider conducting experiments under controlled conditions, reducing the impact of confounding variables that may affect the outcomes.
  2. Make sure to include statistical significance tests to determine the validity of the findings. 
  3. Perform sensitivity analysis to determine the impact of various hyperparameters in the DSBL network, which can help improve the network and better understand how parameter choices affect results.
  4. Replicate the experiments to ensure the robustness of the findings. Replicating the experiments with different datasets or settings can strengthen the reliability of the conclusions.
  5. Include an in-depth error analysis to understand the limitations of the DSBL network. Analyzing the types of errors the network makes can help identify areas for improvement.

The authors should include the results section to effectively convey the research outcomes, enhance the comprehensibility of the work, and facilitate the understanding of the findings.

Comments on the Quality of English Language

Some sentences are long and complex. Consider breaking them into shorter, more concise sentences for better readability.

Use transitional words and phrases (e.g., 'however,' 'therefore,' 'in contrast') to clarify the relationships between ideas.

Author Response

Dear Prof.,

We sincerely appreciate your careful reviewing our manuscript entitled “Deep Unfolding Sparse Bayesian Learning Network for Off-Grid DOA Estimation with Nested Array” [ID: remotesensing-2637039]. Those comments and suggestions are all valuable and quite helpful for improving our paper, as well as the important guiding significance to our researches. We have studied the comments carefully and have made some revisions which we hope meet with approval. 

Reviewer 2 Report

Comments and Suggestions for Authors

This paper proposes a deep unfolding sparse bayesian learning network for off-grid DOA estimation with nested array. Combining on-grid spatial spectrum estimation and off-grid quantization error estimation, the network obtain better performance compared to other exisiting methods.. The method proposed is novel and validated by simulatiion under differrent SNR and number of snapshots.

However,there are some suggestions for improving the paper before it is accepted.

1.The paper should further explain the distinct features of the designed deep network for the nested array compared to other forms, such as uniform array.

2.Some methods in the part "Computer Simulation Experiments" should be included in the "Introduction", such as renference [35],[36],[37];

3.The term "solid circles" in line 86 and  "hollow circles" cannot be found in content of the part;

4.In line 228, before giving the conclusion "the proposed DSBL network can determine the off-grid DOA 228 without training labels and large-scale training data" , there should be a necessary analysis of the reason;

5.The paragraph between line 315 and 319 has no corresponding figures.

Comments on the Quality of English Language

The paper is written well and easy to understand.

Author Response

(The authors gave the same response as above.)

Reviewer 3 Report

Comments and Suggestions for Authors

This paper presents a DSBL network for off-grid DOA estimation by using the geometry of nested array. Compared with the existing methods, the proposed DSBL network achieved more accurate DOA estimation in the cases of limited snapshot numbers and low SNRs. Overall, the paper is well organized, some comments are given as follows.

1. In the simulation, why not provide the results of uniform array to show the advantages of nested array?

2. Compared to the corresponding model-driven method, why not provide the results of computational time to show the advantages of deep unfolding network?

3. How can the authors determine the optimum number of layers in the network?

4. Furthermore, there are some typos that need correction and clarification.

Comments on the Quality of English Language

See the comments above. 

Author Response

(The authors gave the same response as above.)

Round 2

Reviewer 1 Report

Comments and Suggestions for Authors

The authors improved the quality of their contribution. The authors have diligently incorporated all the suggestions, significantly enriching the text.

Comments on the Quality of English Language

Minor editing of English language required